# Leucyl-tRNA Synthetase Inhibitor, D-Norvaline, in Combination with Oxacillin, Is Effective against Methicillin-Resistant *Staphylococcus aureus*

**DOI:** 10.3390/antibiotics11050683

**Published:** 2022-05-18

**Authors:** Hong-Ju Lee, Byungchan Kim, Suhyun Kim, Do-Hyun Cho, Heeju Jung, Wooseong Kim, Yun-Gon Kim, Jae-Seok Kim, Hwang-Soo Joo, Sang-Ho Lee, Yung-Hun Yang

**Affiliations:** 1Department of Biological Engineering, College of Engineering, Konkuk University, Seoul 05029, Korea; oneul1210@naver.com (H.-J.L.); 951022qudcks@naver.com (B.K.); gsm06136@naver.com (S.K.); ehguswh1997@naver.com (D.-H.C.); gda06073@naver.com (H.J.); 2College of Pharmacy and Graduate School of Pharmaceutical Sciences, Ewha Womans University, Seoul 03760, Korea; wooseongkim@ewha.ac.kr; 3Department of Chemical Engineering, Soongsil University, Seoul 06978, Korea; ygkim@ssu.ac.kr; 4Department of Laboratory Medicine, Kangdong Sacred Heart Hospital, Hallym University College of Medicine, Seoul 05355, Korea; jaeseok@hallym.ac.kr; 5Department of Biotechnology, College of Engineering, Duksung Women’s University, Seoul 01369, Korea; hwangsoojoo27@duksung.ac.kr; 6Department of Pharmacy, College of Pharmacy, Jeju National University, Jeju-si 63243, Korea; leesangho@jejunu.ac.kr

**Keywords:** MRSA, D-norvaline, aminoacyl-tRNA synthetase inhibitor, drug combination therapy

## Abstract

Methicillin-resistant *Staphylococcus aureus* (MRSA) is a pathogenic bacterium that causes severe diseases in humans. For decades, MRSA has acquired substantial resistance against conventional antibiotics through regulatory adaptation, thereby posing a challenge for treating MRSA infection. One of the emerging strategies to combat MRSA is the combinatory use of antibacterial agents. Based on the dramatic change in phospholipid fatty acid (PLFA) composition of MRSA in previous results, this study investigated branched-chain amino acid derivatives (precursors of fatty acid synthesis of cell membrane) and discovered the antimicrobial potency of D-norvaline. The compound, which can act synergistically with oxacillin, is among the three leucine-tRNA synthetase inhibitors with high potency to inhibit MRSA cell growth and biofilm formation. PLFA analysis and membrane properties revealed that D-norvaline decreased the overall amount of PLFA, increasing the fluidity and decreasing the hydrophobicity of the bacterial cell membrane. Additionally, we observed genetic differences to explore the response to D-norvaline. Furthermore, deletion mutants and clinically isolated MRSA strains were treated with D-norvaline. The study revealed that D-norvaline, with low concentrations of oxacillin, was effective in killing several MRSA strains. In summary, our findings provide a new combination of aminoacyl-tRNA synthetase inhibitor D-norvaline and oxacillin, which is effective against MRSA.

## 1. Introduction

*Staphylococcus aureus* (*S. aureus*) is a gram-positive bacterial strain, which causes acute and chronic diseases, ranging from dermal infection to sepsis [1,2]. Particularly, treatment of infections caused by methicillin-resistant *S. aureus* (MRSA) is hampered by the species’ ability to adapt and to resist multiple antibiotics, such as penicillin, methicillin, and cefazolin [3,4]. This evolutionary potential has hindered the decades-long effort to combat *S. aureus*-caused diseases and to evade antibiotics resistant mechanisms of MRSA. Moreover, MRSA can be generally classified into healthcare-associated MRSA (HA-MRSA) and community-associated MRSA (CA-MRSA), which are distinguished by the integration of a mobile genetic element, staphylococcal cassette chromosome mec (SCCmec), consisting of *mec* and *ccr* gene complexes encoding recombinase proteins [5]. HA-MRSA carries SCC*mec* type II or III; their *mec* operon contains *mecI* gene repressing *mecA* expression allowing the strains to have relatively higher resistance against beta-lactam antibiotics than the CA-MRSA [6,7]. The various components and arrangements of the genetic elements allow MRSA strains to have a wide range of resistance spectrum. Previous studies have described unique genetic patterns and molecular characteristics in terms of SCCmec types. HA-MRSA shows higher antibiotic susceptibility than CA-MRSA and is believed to be associated with more fatal infections [8]. However, a recent study on CA-MRSA showed high antibiotic resistance with increased virulence, rapidly spreading into the community [9,10]. Therefore, non-antibiotics, such as flavonoids, as well as surfactants, essential oils, and fatty acids combined with conventional antibiotics, have been studied to explore their antibacterial activity [10,11,12]. Aminoacyl-tRNA synthetase (aaRS) inhibitors have been identified as a new class of candidates for drug discovery and development [13].

Previous studies have demonstrated the use of phenotypic features to describe MRSA; this led to the discovery that the cell membrane holds one of the keys to treating MRSA and evading its resistance mechanism [14,15,16]. The cellular fatty acids of staphylococci are known to consist of mainly branched-chain fatty acids (BCFAs) with some straight-chain fatty acids (SCFAs). In bacteria, BCFAs were generated from the catabolism of the branched-chain amino acids (BCAAs), leucine, valine, and isoleucine. Therefore, this study focused on the membrane-dependent changes in MRSA resistance and deduced that the compound which affects the membrane may affect antibiotic resistance in MRSA [17,18,19].

Three leucyl-tRNA synthetase (LeuRS)-targeting compounds—hydroxycinnamic acid (HCA), norvaline, and tribenuron-methyl—were selected to assess their ability to inhibit MRSA growth. HCAs are a class of phenolic compounds characterized by the structural feature of a phenolic ring and radical containing a carboxyl group. They are potential inhibitors targeting the synthetic site of LeuRS [20,21]. Norvaline is a byproduct of the leucine biosynthesis process that binds to the active site of LeuRS as a strong leucine analog. Tribenuron-methyl is a sulfonylurea herbicide to inhibit acetolactate synthase (ALS) in plant and microorganism, which catalyzes the synthesis of BCAAs [22].

We hypothesized that these LeuRS inhibitors can trigger non-cognate leucinylation, thus rearranging the defense mechanism of antimicrobial agents, as aaRS plays an important role in protein biosynthesis by esterifying amino acids with corresponding tRNAs to form aminoacyl-tRNAs. Moreover, amino acid acyl-tRNA synthetase can be an inhibitory target to attenuate cell growth by hindering protein synthesis under infectious conditions [13,23,24,25,26,27].

Therefore, the aim of this study was to examine the possibility of a synergistic inhibitory effect of LeuRS inhibitors (hydroxycinnamic acid (HCA), norvaline, and tribenuron-methyl) with oxacillin against MRSA via combinational therapy. The MRSA USA300 (LAC) strain was used as a model organism of evaluating compounds’ antibacterial properties. Furthermore, deletion mutants and clinically isolated MRSA strains were examined to demonstrate that it is possible to kill other resistant strains via the new combination strategy. By finding the novel potential synergistic anti-MRSA agents, this study hypothesized that the quick spreading multi-drug resistance in MRSA could be delayed.

## 2. Results

### 2.1. Compound Screening Inhibiting Cell Growth of LAC with Oxacillin

Due to the chemical similarity of valine, isoleucine, and leucine, leucine synthetase has been posed with the difficulty of identifying the amino acid to incorporate into proteins [13,24,25,26,27]. Unless mischarged amino acid has been repaired or eliminated by the editing process of synthetase in organisms, it affects their growth and reproduction into the next generation. Moreover, previous studies have revealed that random variation of non-cognate amino acid peptides could lead to mutated internal metabolism, suggesting detrimental or even beneficial evolution [28,29,30].

MIC of oxacillin with three individual LeuRS inhibitors was measured by spectrometry analysis to select the synergistic antimicrobial agent, and the results are shown in (Figure 1). After treating with the three compounds, each combined with 12.5 µg/mL of oxacillin, overall LAC cell growth and biofilm formation decreased compared to the control. In particular, in the LAC group treated with 995.8 µg/mL of DL-norvaline and 12.5 µg/mL of oxacillin the cell growth and biofilm formation were inhibited by 62.83% and 12.4%, respectively, compared to the control group. Moreover, the *S. aureus* LAC strain had decreased cell growth by 0.4-fold, and it rarely produced biofilm. Previous studies have reported that oxacillin MIC of LAC strain is 32 µg/mL; however, we observed the antimicrobial concentration of oxacillin halved with norvaline treatment.

### 2.2. Inhibitory Effect of Norvaline with Oxacillin

To assess the enantiomer-specific activity of norvaline in inhibiting bacterial growth, cell growth and biofilm formation were compared after treating the LAC strain with D-form, L-form, and DL-form of norvaline with and without oxacillin (Figure 2a,b). Both cell growth and biofilm significantly decreased when D-norvaline combined with oxacillin were used. The survival rate and biofilm formation rate of the LAC strain were reduced by 8.9% and 6.2%, respectively, compared to those of the control. This suggested that the inhibitory effect of DL-norvaline in previous results was caused by D-norvaline. Moreover, we conducted a serial dilution of the three forms of norvalines with a 12.5 µg/mL oxacillin mixture to investigate the MIC of norvaline in LAC (Figure 2c). As expected, cell growth sharply decreased in 995.8 µg/mL of D-norvaline and 2048 µg/mL of DL-norvaline. However, the mixture of oxacillin and L-norvaline did not inhibit cell growth at the concentration range of 0–2048 µg/mL.

### 2.3. Effect of D-Norvaline on Phospholipid Fatty Acid Composition and Membrane Properties of LAC

Since amino acid analogs affect the synthesis of phospholipid fatty acids in many microorganisms, we conducted PLFA analysis after D-norvaline treatment and identified the changes in fatty acid composition and distribution (Figure 3).

Gas chromatography–mass spectrometry (GC-MS) analysis showed a 69% decrease in total fatty acid concentration (per g cell) in the D-norvaline-treated LAC strain. However, the ratio of each fatty acid from untreated LAC and D-norvaline-treated LAC was small and made no difference regardless of the presence of D-norvaline (Figure 3a).

We also measured membrane properties such as membrane fluidity and hydrophobicity to evaluate PLFA changes affected by norvaline, which is reported in previous studies [18,31] (Figure 4). D-norvaline addition was observed to increase the membrane fluidity, which controlled LAC. According to previous studies, the strain with high antibiotic resistance was expected to have relatively low membrane fluidity [31]. However, norvaline-treated LACs recorded a higher fluidity than the untreated LAC, even though their resistance to antibiotics was much lower (Figure 4a). These findings contrasted with our previous research, thus reinforcing that there is a considerable number of factors regulating antibiotic resistance. Furthermore, both L- and D-norvaline-treated LAC had decreased membrane hydrophobicity (Figure 4b). This may be due to the difference in total fatty acid concentration via identical controlling cell mass. In summary, our study suggests that D-norvaline indirectly affected the oxacillin resistance of LAC just by diminishing total cell membrane fatty acids. These quantity and quality changes of MRSA membrane caused by D-norvaline could reduce the concentration of oxacillin required.

### 2.4. Different Gene Expression Patterns of LAC with D-Norvaline and CHANGES in Phenotypes

In norvaline-rich conditions, LeuRS confronts a high validity of translational error by incorporating non-cognate amino acids [32]. Therefore, we demonstrated how D-norvaline combined with oxacillin led to a synergistic antimicrobial effect in the transcription level of several genes related to antibiotic resistance or virulence factors in LAC, such as staphlyococcal accessory gene regulator A gene (*sarA*), PBP2a encoding gene (*mecA*), staphyloxanthin synthase gene (c*rtM*), and regulation of exotoxin secretion gene (*saeS*) [33,34] via semi-quantitative RT-PCR (Figure 5a). As one of the housekeeping genes, *gyrB* was expressed identically in the control and 995.8 µg/mL of D-norvaline-treated LAC, while changes were observed in the transcription level of *sarA*, *saeS*, *mecA*, and *crtM*.

The *sarA* gene plays a major role in regulating resistance and virulence in MRSA. Such regulators include accessory gene regulator protein (Agr) and Sae families. Therefore, we investigated the alteration of gene expression under *sarA* regulation by D-norvaline. As previous studies have revealed that β-lactam antibiotics, such as oxacillin reduce the Agr expression to inhibit quorum sensing [35,36], it was expected that D-norvaline attenuates LAC against antibiotics by decreasing *sarA* expression level. Our result showed that the expression levels of *sarA* and *mecA* genes, which encode a penicillin-binding protein, PBP2a, decreased when exposed to D-norvaline, suggesting that D-norvaline acts similarly to oxacillin in protein translation. Similar to D-norvaline, trans-cinnamaldehyde (TCA), which exhibits a synergistic antibacterial effect in combination with conventional antibiotics, showed that transcriptional and relative expression levels of *mecA* and PBP2a were significantly inhibited in a dose-dependent manner [37]. Also, this result corresponds with decreased MIC in LAC with combinational therapies of D-norvaline and oxacillin as an actual phenotype.

In general, a decreased level in these global gene regulators is known to downregulate gene expression of *mecA*, *saeS*, and *crtM* [36]. However, this study observed that the expression levels of *saeS* and *crtM* increased upon exposure to D-norvaline, although *sarA* and *mecA* decreased. Staphyloxanthin (STX) extraction and quantification were examined to confirm whether increased *crtM* expression level correlates with STX production (Appendix A). D-norvaline increased STX contents from 1.36% to 1.84% (STX O.D 470 nm/cell mass O.D 600 nm * 100). In contrast, increased expression level of *saeS* did not lead to increased casein hydrolysis activity (Appendix A). This is because *saeS* gene expression and the activity of D-norvaline as a leucinyl synthesis inhibitor were involved in proteolysis.

### 2.5. D-Norvaline on Deletion Mutant Strains and Clinically Isolated MRSA Strains

To determine the specific affinity to genetic molecules, 995.8 µg/mL of D-norvaline was used along with oxacillin, and the cell growth was measured (Appendix A). The data showed that D-norvaline decreased the growth of Δ*sarA*, Δ*agr*, Δ*codY*, Δ*arlS*, and Δ*rpoF* mutants, except for Δ*saeS* and Δ*mecA* mutants; it also showed decreased oxacillin MIC.

To expand the application, we used clinically isolated MRSA strains, which contain different SCCmec types and characterized the optimal concentration of D-norvaline and oxacillin by checkerboard MIC assay.

The strains were classified into two groups: the CA-MRSA strain carries SCCmec type IV (LAC, MW2, MRSA6203, MRSA7875, and MRSA28985), and the HA-MRSA strain carries SCCmec type II or III (MRSA2065, MRSA6288, MRSA7557, MRSA8471, MRSA9291, MRSA12779 and MRSA14278) (Figure 6). In contrast, the D-norvaline MIC of CA-MRSA was examined with 12.5 µg/mL of oxacillin. Since HA-MRSA could not be killed by low oxacillin concentration, their cell growth was measured by various oxacillin concentrations with a fixed D-norvaline concentration of 995.8 µg/mL. The data is summarized in Table 1. Combination therapy of D-norvaline and oxacillin was effective in lowering oxacillin dose in five CA-MRSA strains (MRSA6230, MRSA7875, MRSA14459, and MRSA28985) carrying SCCmec type IV. Especially, the oxacillin MIC of MRSA14459 showed a significant drop to 12.5 µg/mL compared to its original MIC (1024 µg/mL). Also, a few HA-MRSA carrying SCCmec type II or III displayed positive results in decreasing oxacillin MIC. Moreover, D-norvaline (995.8 µg/mL) depleted oxacillin MIC (1024 to 256 µg/mL) to completely inhibit the growth of SCCmec type III strain MRSA12779. These clinical MRSA strains exhibited a <0.5 fractional inhibitory concentration (FIC) value, which indicates the synergism between D-norvaline and oxacillin. The findings suggest the applicability of D-norvaline and oxacillin combination treatment in further clinical and non-clinical practices.

## 3. Discussion

Emergence of multi-drug resistance in MRSA has required more complex and sophisticated treatment for decades. Combination therapy can be an effective tool for dealing with the growing problems of antibiotic abuse and the rapidly evolving resistance mechanisms of MRSA, enabling doctors to treat both clinical and non-clinical MRSA infections. This study demonstrated that D-norvaline, in combination with oxacillin, inhibited the growth and biofilm formation of MRSA strains. D-norvaline significantly decreased the MICs of oxacillin in many MRSA strains, including CA-MRSA and HA-MRSA.

To investigate the antibacterial activities of D-norvaline, we observed the changes in phospholipid fatty acid. PLFA results revealed that the total fatty acid quantity decreased substantially after treating with 995.8 µg/mL of D-norvaline. Moreover, the changes in membrane fluidity and hydrophobicity followed by the reduction in PLFA quantity were also measured. This finding may indicate that as a branched-chain amino acid substrate, D-norvaline hindered the fatty acid synthesis pathway and facilitated oxacillin permeation inside the cells.

Additionally, quantitative RT-PCR was performed to compare mRNA expression levels of *sarA*, *mecA*, *saeS*, and *crtM* genes. Expression levels of *sarA* and *mecA* decreased, whereas those of *saeS* and *crtM* increased after D-norvaline treatment. Although, the changes in gene expressions and fatty acid synthesis may not be directly linked, those controlling D-norvaline could elucidate how it lowers oxacillin MIC in MRSA strain. The *sarA* gene is responsible for regulating resistance and virulence factors in MRSA, especially by activating the Agr system; the expression of *mecA* gene is downregulated and becomes less resistant to oxacillin. Simultaneously, the increase in both *saeS* and *crtM* gene expression levels and staphyloxanthin production suggested that D-norvaline positively regulates the virulence factors.

Because of the complexity of two combined compounds, it was hard to conclude the accurate mechanism of the antibacterial function. However, this study revealed extraordinary features in MRSA cell membrane and gene expression after treating D-norvaline.

## 4. Materials and Methods

### 4.1. S. aureus Strain, Media, and Culture Condition

To confirm antibiotics resistance in MRSA strain, *Staphylococcus aureus* strain USA300-0114 (LAC) (CDC 2003) was prepared after pre-culture in liquid tryptic soybean broth (TSB). A single colony of the LAC strain on the TSB 1.5% (*w*/*v*) agar plate was inoculated with 5 mL of TSB medium and grown overnight in a shaking incubator at 37 °C (200 rpm). Then, 1% (*v*/*v*) of the cell pre-cultured suspension was inoculated in TSB for subsequent cell cultivation at the same culture condition. For 96-well plate culture, cell cultivation was conducted without shaking.

Deletion mutant (Δ*sarA*, Δ*agr*, Δ*codY*, Δ*arlS*, Δ*rpoF*, Δ*saeS*, Δ*mecA* Tn mutants) strains were obtained from the Department of Laboratory Medicine, Kangdong Sacred Heart Hospital, Hallym University College of Medicine (Seoul, Korea).

The clinically isolated MRSA strains (MRSA2065, MRSA6230, MRSA6288, MRSA7557, MRSA7875, MRSA8471, MRSA9291, MRSA12779, MRSA14459, and MRSA28985) were prepared. The type and MIC of these strains are listed in Table 1.

### 4.2. Minimum Inhibitory Concentration (MIC) and Biofilm Formation Test

To investigate the MIC and biofilm formation of antibiotics, 200 μL of TSB containing serial diluted oxacillin was prepared in a 96-well plate. A total of 1% (*v*/*v*) of the pre-cultured cell was inoculated and suspended in each 96-well plate and incubated at 37 °C for 24 h without shaking. For cell growth measurement, optical density was observed using a 96-well plate reader (Molecular Devices, San Jose, CA, USA), and absorbance was read in 600 nm wavelength. Biofilm formation was analyzed using the crystal violet staining method following a previous report [17]. Biofilm fixation was performed by eliminating the supernatant and replacing it with methanol. Then the plate was air-dried for further staining with 0.2% crystal violet dissolved in 20% methanol. After 5 min of treatment, the remaining dye was gently removed with distilled water (DW), and absorbance was measured at 600 nm [38].

### 4.3. Screening of Inhibitory Compound in LAC Strain

Based on the chemical and structural potential to bind to LeuRS, three aaRS inhibitors, hydroxycinnamic acid, DL-norvaline, and tribenuron-methyl, were selected to investigate the synergistic inhibitory effect with oxacillin on LAC strain. Three compounds were combined with oxacillin in 1941.8 µg/mL (10 mM), 995.8 µg/mL (8.5 mM), and 39.5 µg/mL (0.1 mM), respectively. To avoid complete inhibition of cell growth by oxacillin, we treated LAC strain with 12.5 µg/mL of oxacillin as a subinhibitory concentration, which is lower than 0.5 MIC (16 µg/mL) of LAC strain. A 200 μL mixture of oxacillin and the compounds in TSB were prepared in a 96-well plate, and 1% (*v*/*v*) of pre-cultured LAC strain was inoculated. Culture condition analysis methods were the same as previously described.

### 4.4. Phospholipid Fatty Acid (PLFA) Analysis

To see the changes in phospholipid fatty acid, 100 mL of the liquid culture was performed in TSB with 1% (*v*/*v*) inoculum of LAC strain (WT) with and without 995.8 µg/mL of D-norvaline at 37 °C at 200 rpm for 24 h. The cell pellet was collected by centrifugation at 3000× *g* for 15 min after 24 h of cultivation and suspended with 0.15 M citric acid buffer: chloroform: methanol in 1:1:1 (*v*/*v*). The samples were incubated at 37 °C in a shaking incubator (200 rpm) for 2 h to extract total fatty acids. To avoid oxidation, the chloroform phase was transferred into a glass vial by light centrifugation and evaporated at 50 °C in a heat block under compressed nitrogen gas (N_2_). Chloroform (2 mL) was added into dried vials and vortexed for 1 min. A sialic column was prepared by washing with 5 mL of acetone and 5 mL of chloroform serially. The samples were loaded into the column, and elution was performed with 5 mL each of chloroform, acetone, and methanol. The methanol phase was collected for further PLFA analysis. Mild alkaline methylation was conducted for gas chromatography–mass spectrometry (GC-MS) analysis. A total of 1 mL of toluene/methanol (1:1, *v*/*v*) and 0.2 M of KOH/methanol were added to the methanol samples at 37 °C for 15 min. After cooling to room temperature, 2 mL of 4:1 n-hexane/chloroform, 0.3 mL of 1 M acetic acid, and 2 mL of distilled water were added to the sample. The upper hexane layer was collected into another glass vial and concentrated under compressed N_2_. This step was repeated twice with fresh 2 mL of 4:1 n-hexane/chloroform. The fatty acids were dissolved into 1 mL of chloroform and analyzed using GC-MS [19].

The GC-MS used was a Perkin Elmer Clarus 500 gas chromatograph connected with a Clarus 5Q8S mass spectrometer at 70 eV (*m*/*z* 50–500; source at 230 °C and quadruple at 150 °C) in electron ionization mode with an Elite 5 MS capillary column (30 m × 0.32 mm × 0.25 μM film thickness; PerkinElmer, Waltham, MA, USA). As the carrier gas, Helium was used at a flow rate of 1.0 mL/min. The inlet temperature was maintained at 300 °C, and the oven temperature was controlled to start at 120 °C for 5 min before increasing to 300 °C at a rate of 4 °C/min and then maintained for 20 min. The sample injection volume was 1 μL, with a split ratio of 50:1. The analytical standards for each fatty acid were identified in our previous study [39].

### 4.5. Analysis of Membrane Hydrophobicity and Fluidity

Membrane properties play an important role in maintaining cell structure and by preventing antibiotics permeation into the cytoplasm. To determine the effects of norvaline on the cell membrane, we conducted membrane hydrophobicity and fluidity tests.

Membrane hydrophobicity was tested based on the difference in adsorption between the hydrophobic cell surface and octane [39,40]. The cells were harvested by centrifugation (3000× *g*, 10 min) and resuspended in cold 0.8% saline, adjusting the optical density to 0.6 at 595 nm. N-octane (0.6 mL) was added to aliquots of the suspension (3 mL). The suspensions were vortexed for 2 min, then allowed to stand to form two layers (n-octane and saline) separation for 15 min. Then, the decrease in turbidity of the saline phase was calculated.

Membrane fluidity analysis was conducted using a fluorescent probe that reacts with polarized light in a membrane, producing fluorescence polarization, resulting in a measurable polarization value [39]. The samples were washed twice in saline (pH 7.0) and resuspended at a concentration of 1 × 108 cells/mL. Thereafter, 0.2 μM of 1,6-diphenyl-1,3,5-hexatriene (DPH; supplied by Life Technologies, Carlsbad, CA, USA, 0.2 mM stock solution in tetrahydrofuran) was immediately added, and the sample was incubated at 37 °C for 30 min. Fluorescence polarization values were determined using a Synergy 2 Multi-Mode microplate reader (BioTek, Winooski, VT, USA) with sterile black flat-bottom Nunclon Delta-Treated 96-well plates (Thermo Fisher Scientific, Waltham, MA, USA). The fluorescence filters used were a 360/40 nm polarized excitation filter set in a vertical position and a 460/40 nm polarized emission filter (BioTek) set either in a vertical (IVV) or horizontal (IVH) position.

### 4.6. Quantification of Staphyloxanthin (STX)

Cells were grown in 5 mL of TSB with and without 995.8 µg/mL of norvaline at 37 °C and 200 rpm for 24 h and harvested by centrifugation (3000× *g*, 15 min). To quantify total STX from identical cell mass of each sample, the cell pellets were suspended using phosphate-buffered saline (PBS) and adjusted into the same amount by measuring optical density at 600 nm. As the same mass of cells was collected, the cells were washed with 2 mL of PBS buffer, and the supernatant was removed. The pellet was resuspended to 500 mL of methanol and incubated at 55 °C for 20 min. Following centrifugation, 200 μL of methanol-containing pigment was obtained. The pigment content of each sample was determined by optical density at 470 nm via plate reader spectrometer [5].

### 4.7. Semi-Quantitative RT-PCR

LAC strain was pre-cultured in 5 mL of TSB media, and 1% (*v*/*v*) of media was inoculated into 5 mL of TSB with and without 995.8 µg/mL of D-norvaline. The culture was conducted overnight at 37 °C in a shaking incubator (200 rpm). After incubation, 2 mL of media was centrifuged at 3000× *g* for 15 min to remove supernatant. Then, total RNA was prepared using RNeasy Mini Kit (Qiagen, Hilden, German) and reverse transcriptase (Invitrogen Co., Carlsbad, CA, USA) to synthesize the cDNA. The primers used in this study were designed from NCBI (Primer-BLAST, Available online: https://www.ncbi.nlm.nih.gov/tools/primer-blast/ (accessed on 10 April 2022)), and these primers can synthesize up to 200 bp of PCR product for the comparison of gene expression levels. To optimize PCR cycle number, gyrB gene expression level was monitored by different cycles; 25 cycles were used for further analysis. In manual methods, semi-quantitative RT-PCR was conducted using LA Taq with GC buffer I (Takara Medical Co. Ltd., Seoul, Korea) [38].

### 4.8. Quantitative Reverse Transcription PCR (RT-qPCR)

The log fold changes of the target genes were measured to gain a deeper insight into gene expression correlated with phenotypic changes. We used the set of cDNA as a template DNA which was synthesized from isolated RNA of *S. aureus* LAC strain, with or without D-norvaline treatment. The reactions were performed using TOPreal^TM^ qPCR 2X PreMIX (SYBR Green with high ROX) (Enzynomics, Daejeon, Korea). DNA gyrase subunit B (gyrB) was used to normalize gene expression levels.

### 4.9. Plate-Based Motility and Protease Activity Test

To determine the effect of D-norvaline on motility, soft agar assay was conducted as previously reported [17]. Pre-cultured LAC strain was spotted at 2 μL on the center of 0.24% tryptic soy agar (TSA) plates. After 10 min, 2 μL of 995.8 µg/mL D-norvaline solution was also dropped on the same spot. Then, the plates were incubated overnight at 37 °C. The motility of the LAC strain was analyzed by measuring the length of colony diameter after incubation.

Protease activity was also measured on agar plate assay. TSA with 2.8 g/L of skim milk powder plate was prepared, and a sterile paper disc was put on the center of the plate. Pre-cultured LAC strain (10 µL) with 10 µL of norvaline was applied on the paper disc, and the plates were incubated overnight at 37 °C. The size of the clear zone around the paper disc was measured.

### 4.10. Checkerboard MIC Assay

To investigate the optimal inhibitory concentration of two reagents, D-norvaline and oxacillin, we conducted a checkerboard MIC assay using a 96-well plate and VIAFLO ASSIST (Bio-Medical Science Co., Ltd., Seoul, Korea). A total of 200 µL of TSB containing serial diluted oxacillin (0–2048 µg/mL) was dispensed over the 96-well plate, and 0–2048 µg/mL D-norvaline was diluted. After inoculating 1% (*v*/*v*) of MRSA strain in pre-cultured TSB, the sample was incubated overnight at 37 °C without shaking, and cell growth was measured by a 96-well plate reader (TECAN, Männedorf, Switzerland) at a 600-nm wavelength.

## 5. Conclusions

This study explored the new synergistic antibacterial effect of D-norvaline with oxacillin with multi-genotype and phenotype data, including gene expressions, antibacterial activities, phospholipid fatty acid analysis, and pigment quantification. Results suggested that oxacillin dosage required to inhibit several MRSA strains can be reduced by the effective application of an antibacterial aaRS inhibitor. This finding will contribute to solving the challenges of treating MRSA infection by using combinational therapy. However, further investigations on possibility and dosage for application in animal models are required.

## Figures and Tables

**Figure 1 antibiotics-11-00683-f001:**
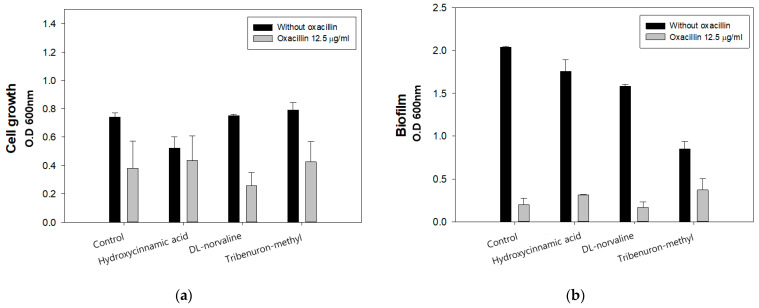
(**a**) Synergistic inhibitory effect of three leucine synthetase (LeuRS) inhibitors on cell growth; (**b**) biofilm formation. (oxacillin 12.5 µg/mL, hydroxycinnamic acid 1941.8 µg/mL (10 mM), DL-norvaline 995.8 µg/mL, tribenuron-methyl 39.5 µg/mL (0.1 mM)). The error bars represent the standard deviation of three replicates.

**Figure 2 antibiotics-11-00683-f002:**
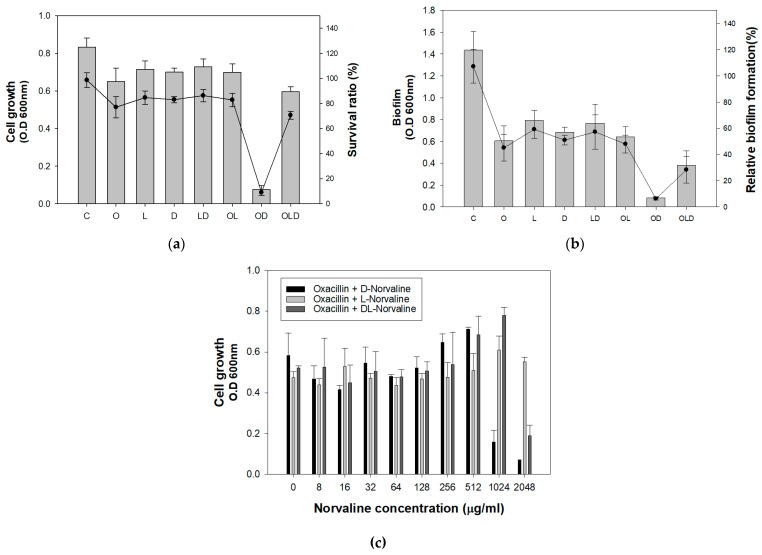
Inhibitory effect of D-norvaline and oxacillin on cell growth and biofilm formation of MRSA LAC strain. (**a**) Cell growth. (**b**) Biofilm formation. (C = control, O = oxacillin 12.5 µg/mL, L = L-norvaline 995.8 µg/mL, D = D-norvaline 995.8 µg/mL, LD = DL-norvaline 995.8 µg/mL, OL = oxacillin 12.5 µg/mL with L-norvaline 995.8 µg/mL, OD = oxacillin 12.5 µg/mL with D-norvaline 995.8 µg/mL, OLD = oxacillin 12.5 µg/mL with DL-norvaline 995.8 µg/mL). (**c**) MIC of norvaline via serial dilution. The error bars represent the standard deviation of four replicates.

**Figure 3 antibiotics-11-00683-f003:**
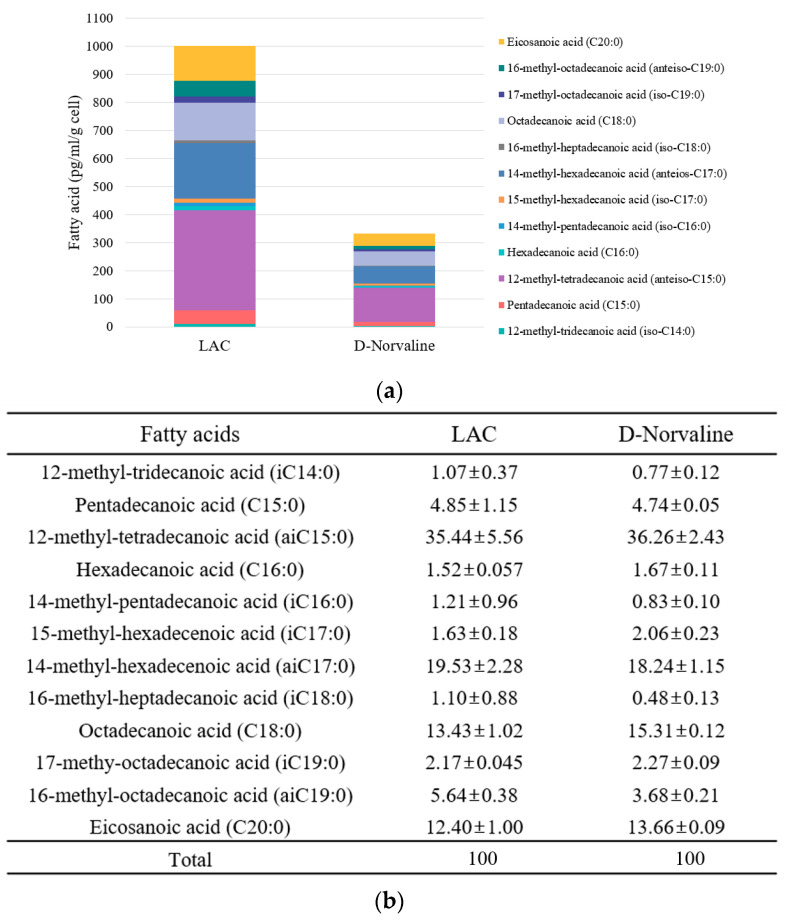
Phospholipid fatty acid analysis of LAC and 995.8 µg/mL of D-norvaline-treated LAC (**a**) Quantification of total fatty acid; (**b**) fatty acid composition ratio (%).

**Figure 4 antibiotics-11-00683-f004:**
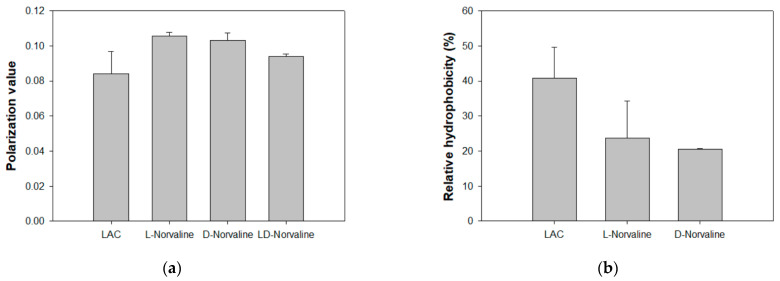
Cell membrane properties with norvaline and oxacillin. (**a**) Membrane fluidity. (**b**) Membrane hydrophobicity. (C = control, TSB, L = L-norvaline 995.8 µg/mL, D = D-norvaline 995.8 µg/mL).

**Figure 5 antibiotics-11-00683-f005:**
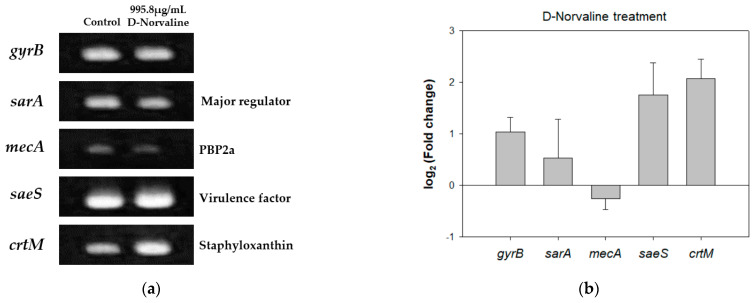
(**a**) Semi-quantitative RT-PCR; (**b**) fold change of *gryB*, *sarA*, *mecA*, *saeS*, and *crtM* gene expressions by RT-qPCR in LAC with and without 995.8 µg/mL D-norvaline. The error bars represent the standard deviation of three replicates.

**Figure 6 antibiotics-11-00683-f006:**
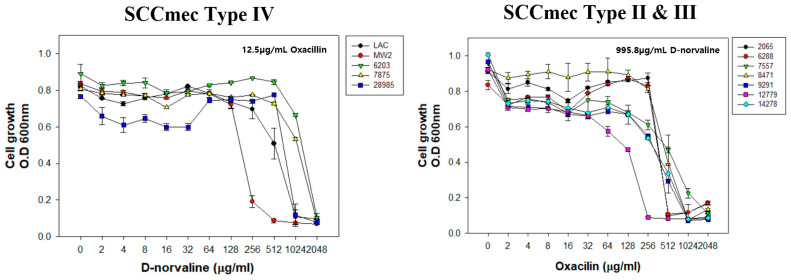
Changes of oxacillin minimum inhibitory concentration (MIC) with D-norvaline treatment in various MRSA strains.

**Table 1 antibiotics-11-00683-t001:** Changes of oxacillin minimum inhibitory concentration (MIC) with D-norvaline treatment in various MRSA strains.

Name	Type	*SCCmec* Type	Oxacillin MIC (µg/mL)		Spa Type	MLST (ST)
			Without D-Norvaline	With 8.5 mM D-Norvaline		
LAC	MRSA	IV	32	12.5	t008	8
MW2	MRSA	IV	32	12.5	t131	1
2065	MRSA	III	1024	512	t037	239
6230	MRSA	IV	128	32	t324	72
6288	MRSA	III	1024	512	t037	239
7557	MRSA	II	1024	1024	t9353	5
7875	MRSA	IV	128	32	t664	72
8471	MRSA	II	1024	1024	t9353	5
9291	MRSA	II	1024	1024	t601	5
12779	MRSA	II	1024	256	t2460	5
14278	MRSA	II	1024	1024	t9353	5
14459	MRSA	IV	1024	12.5	t324	72
28985	MRSA	IV	64	12.5	-	30

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
