# Peer review of "Leucyl-tRNA Synthetase Inhibitor, D-Norvaline, in Combination with Oxacillin, Is Effective against Methicillin-Resistant Staphylococcus aureus"

_antibiotics, 2022, doi:10.3390/antibiotics11050683_

Round 1

Reviewer 1 Report

LAC group was treated with mixture  contained either 995.8 ug/ml of norvaline or 12.5 ug/ml of oxacillin - why these disproportionate amounts were used? There are large amount of norvaline, smaal of oxacillin - is it the same number of moles? Explain. In experimental part, authors described estimation of the "optimal" inhibitory concentration of both reagents, but it is not clear, there is lack of explanation.

There is a lot of abbreviations in the text. PIt is adviced to provide a list of them.

Reviewer 2 Report

Dear Authors

MRSA control is an important issue within healthcare facilities and for community health, this is a very relevant research area, even more when is about improving treatment options.  Efforts made by authors are greatly appreciated, nonetheless I have some questions and I would like to make some observations on the overall information written within the manuscript.

  1. In the introduction, paragraph within lines 60-64 seems unconnected with the rest of the ideas, although later in the text is clearer how membrane status and leucyl-synthetase activity are interconnected, this part of the text could be improved. Perhaps by moving paragraph in lines 136-141 into the introduction, maybe it could gave a clearer idea.
  2. In section 2.1, how concentrations for the inhibitors were selected? 
  3. Norvaline was the compound selected to be used for further assays and is the only which concentration is expressed in μg/mL, why? although in figure 5a its concentration is expressed as 8.5mM.
  4. Figure 1 shows preliminary synergy experiment, however it is not specified how controls were prepared. Are the controls LAC-cultures treated only with the inhibitors? only with oxacillin? or are untreated cultures?. Why each control has different levels of cell growth and biofilm formation? This could be relevant since tribenuron-methyl also seems to significantly reduce oxacillin's concentration.
  5. In line 105 authors stated that MIC of oxacillin has been reported as 32 µg/mL, but in material and methods, it is stablished that 0.5MIC (12,5 µgr/ml) was used "to avoid complete inhibition of cell growth". Which one is the MIC value for oxacillin: 32 or 25 µg/ml? From which reference came the value of 25 µg/ml?
  6. In Figure 2a is evident that mixture of D-norvaline + oxacillin has the highest growth inhibition, nonetheless, in Figure 2c, where serial dilution were performed, at some concentrations the L-norvaline + oxacilline mixture seems more effective (at 256 and 512 µg/ml for instance). How can this phenomenon be explain?
  7.  In Figure 3 authors report a decrease in total fatty acid but no difference in the ratio of each fatty acid. How authors can assure that the observations are not due to less biomass in treated cultures? this is to say, does D-norvaline treated cultures have less cells?
  8. Is there a typo on figure 3a? units are expressed as µµg/ml/g cell, should it be pg/ml/g cell?
  9. There is a typo on the title of Table 1, units are expressed as ug/ml instead of µg/ml
  10. Could values in figure 6 be used for the estimation of Fractional Inhibitory Concentration (FIC)? This estimation could confirm the synergy between D-norvaline and oxacillin.
  11. Phrase in line 247 in the discussion section seems incomplete, "To investigate the reason for the drug." What reason? for what?
  12. The affirmation in line 255 seems risky, based on the results, these are trending observations, but no mechanism was proven, hence any asseveration should be taken carefully. 
  13. References 9 and 25 are duplicated
  14. References 11 and 18 are duplicated
